# Death receptor 6 does not regulate axon degeneration and Schwann cell injury responses during Wallerian degeneration

Bogdan Beirowski*, Haoran Huang, Elisabetta Babetto*

Department of Neurology, The Neuroscience Research Institute, College of Medicine, The Ohio State University Wexner Medical Center, Columbus, United States

## eLife Assessment

In this **valuable** study, through carefully executed and rigorously controlled experiments, the authors challenged a previously reported role of the Death Receptor 6 (DR6/Tnfrsf21) in Wallerian degeneration (WD). Using two DR6 knockout mouse lines and multiple WD assays, both *in vitro* and *in vivo*, the authors provided **convincing** evidence that loss of DR6 in mice does not protect peripheral axons from WD after injury, at least in the specific contexts of the mice and analyses performed in this study. Due to the lack of certain specific parameters from previous studies (sex, age, mouse strains etc.), the exact reasons underlying the observed inconsistencies between current and previous reports on the protective effects of DR6 remains to be determined. Overall, this is a carefully executed study providing invaluable information toward understanding DR6's role (or lack thereof) in axon degeneration.

*For correspondence:
bogdan.beirowski@osumc.edu
(BB);
elisabetta.babetto@osumc.edu
(EB)

**Competing interest:** The authors declare that no competing interests exist.

**Abstract** Axon degeneration (AxD), accompanied by glial remodeling, is a pathological hallmark of many neurodegenerative diseases, leading to the disruption of neuronal connectivity. Understanding the mechanisms in neurons and glia that regulate AxD is essential for developing therapeutic strategies to prevent or mitigate axon loss. Wallerian degeneration (WD) is a well-established model to study the mechanisms of nerve injury-induced AxD, glial responses, and axon-glia interactions. We recently showed that Schwann cells (SCs), the axon-associated glia of the peripheral nervous system, exert protective effects on axons through their rapid metabolic injury response. Enhancing this SC response promotes axon protection during WD. A prior study reported that eliminating the orphan tumor necrosis factor receptor DR6 (death receptor 6, encoded by *Tnfrsf21*) strongly delays AxD and alters SC injury responses during WD, suggesting a possible intersection with our findings. Here, we rigorously revisit the role of DR6 in WD using two independent DR6 knockout mouse lines including the same model used in the previous study. Surprisingly, in contrast to the earlier report, we observed no impact of DR6 deletion on AxD kinetics or SC injury responses across a range of WD assays. Moreover, injured axons in primary neuronal cultures lacking DR6 degenerated at a similar rate as wild-type axons. We conclude that DR6 is dispensable for the regulation of AxD and glial nerve injury responses during WD. Our data argue that any therapeutic benefit from DR6 suppression in neurodegeneration models occurs through mechanisms independent of WD.

## Introduction

Axon degeneration (AxD) is a fundamental pathological process that significantly contributes to the progression of many neurodegenerative disorders such as peripheral neuropathies, amyotrophic

lateral sclerosis, and multiple sclerosis (*Coleman and Höke, 2020*; *Krauss et al., 2020*; *Wilson et al., 2023*; *Conforti et al., 2014*). Wallerian degeneration (WD), the stereotyped breakdown of axons in an injured nerve, has been established as a powerful and well-characterized model for dissecting the cellular and molecular mechanisms governing AxD. While previously regarded as a passive deterioration of compromised axons, AxD is now understood to be an active and tightly regulated process. This process is driven by intrinsic neuronal mechanisms, but recent evidence suggests that it can be modulated by interactions with surrounding glial cells.

Current consensus holds that AxD during WD, and likely in disease contexts, is largely driven by a conserved axonal self-destruction program. Central to this program is the proteasomal degradation of the metabolic enzyme nicotinamide mononucleotide adenylyl transferase 2 (Nmnat2), regulated by the large, multidomain E3 ubiquitin ligasePhr1/Mycbp2 in conjunction with other ubiquitin-proteasome system components, which in turn triggers the activation of the nicotinamide adenine dinucleotide (NAD⁺) hydrolase sterile-α- and TIR motif-containing protein 1 (SARM1) (*Coleman and Höke, 2020*; *Krauss et al., 2020*). This cascade converges on the collapse of axonal metabolism, ultimately triggering rapid structural disintegration of the axon. In parallel, Schwann cells (SCs), the axon-associated glia in the peripheral nervous system, play a non-cell-autonomous role in modulating AxD following injury (*Babetto et al., 2020*; *Mutschler et al., 2023*). Upon axonal damage, SCs adopt a repair-supportive phonotype (*Jessen and Mirsky, 2019*; *Arthur-Farraj et al., 2012*) and undergo rapid metabolic rewiring characterized by a glycolytic shift that antagonizes AxD through metabolic axon-glia crosstalk (*Babetto et al., 2020*). This suggests that axon-glia interactions are critical determinants of degeneration kinetics and may represent therapeutic targets.

Against this backdrop, identifying molecular regulators that influence AxD together with glial injury responses has become an area of particular interest. Death receptor 6 (DR6), a member of the tumor necrosis factor receptor superfamily encoded by *Tnfrsf21*, stands out as a candidate that meets these criteria. A previous study reported that DR6 inactivation in mice dramatically delays WD, preserving disconnected axons for extended periods, and also alters SC behavior (*Gamage et al., 2017*). These findings raised the possibility that DR6 may function at the interface of the axonal self-destruction pathway and glia-mediated axon protection. However, the role of DR6 in WD remains controversial, with limited independent validation. Clarifying whether DR6 is a bona fide component of the WD pathway is therefore essential, not only for understanding the underlying biology but also for evaluating its potential as a therapeutic target for axon protection.

In this study, we re-examine the role of DR6 in AxD and SC injury responses during WD using two independent knockout mouse models and complementary *in vivo* and *in vitro* approaches. Contrary to the prior report (*Gamage et al., 2017*), our findings demonstrate that loss of DR6 does not alter the kinetics of AxD or the characteristic glial response to nerve injury. These results indicate that DR6 is dispensable for WD regulation and suggest that previously reported neuroprotective effects of DR6 suppression likely arise through mechanisms independent of WD.

## Results

### AxD following nerve injury proceeds normally in mice lacking DR6

DR6 has been implicated in axon degeneration (AxD) mechanisms in several neurodegeneration models (*Hu et al., 2013*; *Huang et al., 2013*; *Mishra et al., 2020*; *Wang et al., 2015*). Given the proposed mechanistic overlap with Wallerian degeneration (WD) (*Conforti et al., 2014*), we first sought to confirm that DR6 loss protects axons after injury (*Gamage et al., 2017*). The previous work reported that disconnected axons from DR6 null mice (*Tnfrsf21^{ΔEx2-3/ΔEx2-3}*) are preserved for up to 4 weeks after nerve injury, with a phenotypic penetrance in approximately 38.5% of the animals studied (*Gamage et al., 2017*). Quantitative real-time (qRT) PCR and western blot analysis validated successful disruption of DR6 expression in these constitutive DR6 null mice, and in another DR6 knockout model with a distinct targeting strategy in which we deleted exon 2 of the *Tnfrsf21* gene in all tissues, including germ cells through Cre-mediated recombination (*Tnfrsf21^{ΔEx2/ΔEx2}*) (*Appendix 1—figure 1*). Unlike earlier studies using DR6 null mutants (*Gamage et al., 2017*; *Colombo et al., 2018*), we found no significant abnormalities in early developmental myelin formation at postnatal day 1 (*Appendix 1—figure 2*), or in g-ratio and axon and fiber caliber distributions as assessed by nerve histomorphometry of sciatic and tibial nerve axons from adult *Tnfrsf21^{ΔEx2-3/ΔEx2-3}* and *Tnfrsf21^{ΔEx2/ΔEx2}*

animals (*Appendix 1—figure 3*). This ensures a normal baseline for our comparative WD studies that follow.

We transected sciatic nerves of overall 38 *Tnfrsf21^{ΔEx2-3/ΔEx2-3}* and 18 *Tnfrsf21^{ΔEx2/ΔEx2}* mice and assessed axon survival in the distal nerve stump 3 days post-axotomy by light and electron microscopy in comparison to littermate controls. Surprisingly, we found that all studied DR6 null mice consistently exhibited an almost complete breakdown of axonal structure, identical to control littermate mice (*Figure 1*). In contrast, nerve stumps from mice expressing one copy of the *Wld^S* gene (which substitutes for the loss of Nmnat2 in transected axons), and mice with loss of the pro-degenerative molecules Phr1/Mycbp2 or SARM1 displayed profound preservation of the vast majority of axons, in line with previous work (*Figure 1*).

To detect a potentially smaller delay of AxD in the DR6 null mouse models, we next performed lower stringency assays and studied axon survival 30 hr after nerve injury. This is a time point at which many disconnected axons just start to disintegrate, and ~50% of myelinated axons in tibial nerves remain structurally intact in wild-type mice as classified on high-resolution light and electron micrographs (*Babetto et al., 2020*). Fiber quantification in axotomized nerve segments from *Tnfrsf21^{ΔEx2-3/ΔEx2-3}* and *Tnfrsf21^{ΔEx2/ΔEx2}* mutants demonstrated that the axon survival was statistically indistinguishable from control samples (*Figure 1—figure supplement 1A*). By contrast, Wld^S mice, and mutants lacking Phr1/Mycbp2 or SARM1, displayed significantly enhanced axon survival (*Figure 1—figure supplement 1B*).

We next asked if disconnected nerve stumps from the DR6 null mutants show reduced site-specific phosphorylation (Thr183/Tyr185) and thus activation of c-Jun N-terminal kinases (JNK), as reported previously (*Gamage et al., 2017*). Reduced activation of JNKs, members of the mitogen-activated protein kinase (MAPK) family, has also been observed in axotomized optic nerves under conditions of axon protection conferred by SARM1 loss or expression of a Wld^S variant (*Yang et al., 2015*). In accord with the absence of any structural axon protection, we found that the JNK activation in axotomized nerves from all mutants was indistinguishable from control preparations (*Figure 1—figure supplement 2*).

Together, these data indicate that the loss of DR6 *in vivo* has no obvious effect on the progression of AxD in axotomized nerves.

## Normal Schwann cell injury responses and myelin remodeling dynamics during WD in the absence of DR6

Schwann cells (SCs) rapidly transdifferentiate and dismantle their myelin sheaths during WD, a process that is largely orchestrated by the transcription factor c-Jun (*Jessen and Mirsky, 2019*; *Arthur-Farraj et al., 2012*). The deletion of *Tnfrsf21* has been claimed to alter the typical SC myelin reorganization observed after nerve lesion (*Gamage et al., 2017*), suggesting abnormal SC transdifferentiation. To determine if DR6-deficient SCs show abnormal reactions to nerve injury, we analyzed c-Jun expression by immunofluorescence on sciatic nerve cross sections distal to the site of nerve injury (*Figure 2*). Additionally, we examined the ultrastructure of dedifferentiated SCs (*Figure 2—figure supplement 1A*) and quantified the area occupied by myelin sheaths and myelin debris profiles on osmium tetroxide- and toluidine blue-stained nerve sections (*Figure 2—figure supplement 1B and C*). We did not detect any significant differences in any of these assays at any time point investigated in the samples from DR6 null mutants as compared to control preparations (*Figure 2*, *Figure 2—figure supplement 1*). These data demonstrate that DR6-deficient SCs show normal injury responses following nerve lesion.

## Injured DRG neurites in primary neuronal cultures from DR6 knockout mice degenerate rapidly

The results above contrast sharply with the previously reported dramatically delayed AxD during WD in DR6 null mice (*Gamage et al., 2017*). In an attempt to reconcile the contradictory findings, we finally considered the possibility that more subtle changes in the stability of injured axons could be potentially unveiled *in vitro* using embryonic neuronal explant cultures from the DR6 mutant models. Such assays were previously extensively used to reveal and characterize axonal DR6 phenotypes, although a key publication in this regard was recently retracted (*Nikolaev et al., 2024*). Importantly,

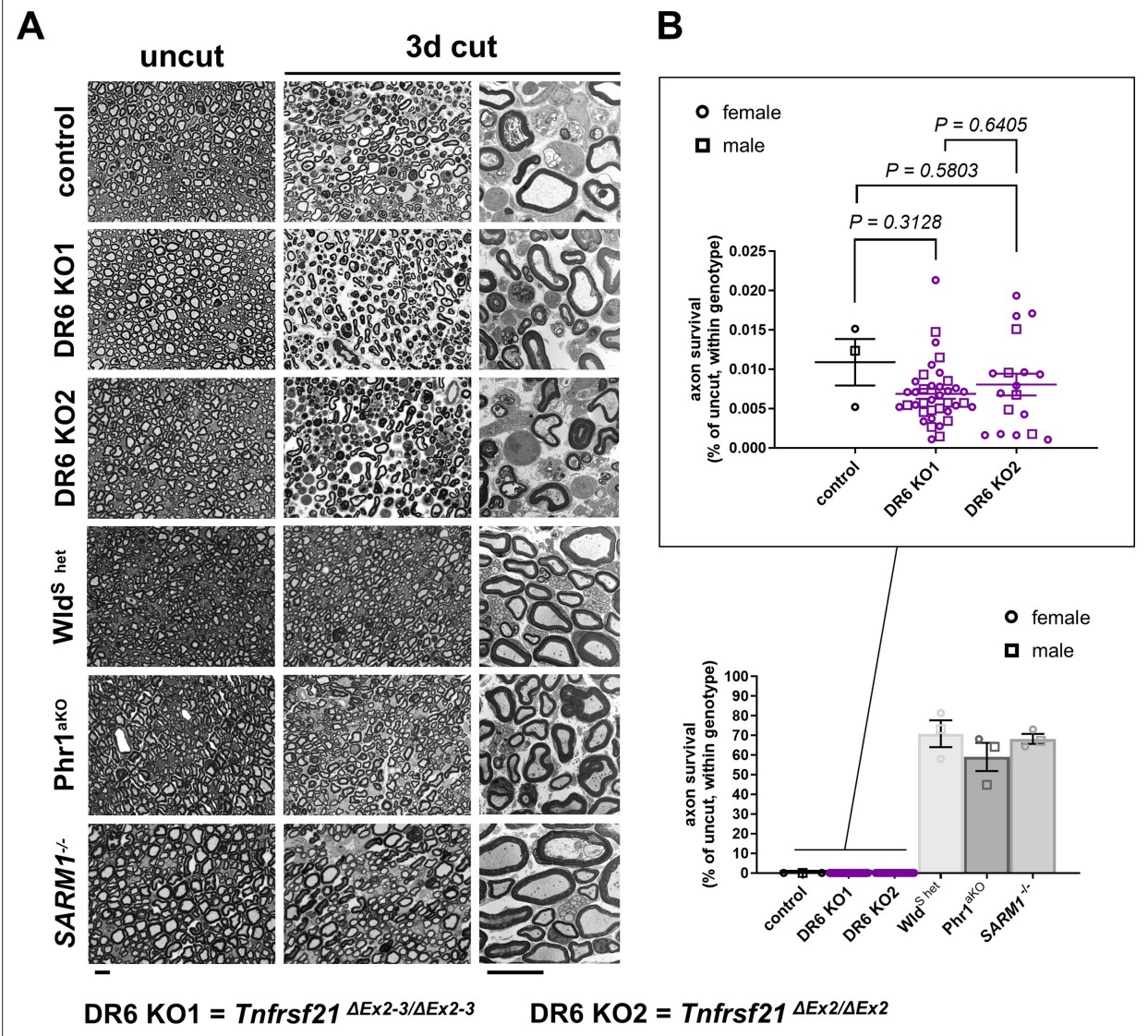

**Figure 1.** Rapid Wallerian degeneration in mice lacking DR6 in comparison to multiple mutants with delayed degeneration, assessed at 3 days after axotomy. (**A**) Representative semithin (first and second columns) and electron micrographs (third column) of transverse sciatic nerve sections of distal nerve stumps from control and the indicated mutant mice 3 days after sciatic nerve transection. Note the complete structural disintegration of transected axons with absent or floccular cytoskeleton and collapsed myelin sheaths in the preparations from control, *Tnfrsf21*$^{\Delta Ex2-3/\Delta Ex2-3}$, and *Tnfrsf21*$^{\Delta Ex2/\Delta Ex2}$ mice. In contrast, the majority of disconnected axons from heterozygous Wld$^S$ mice, and from Phr1 and SARM1 knockout mice, are structurally preserved with uniform cytoskeleton and intact myelin sheaths. Scale bars: 10 µm. (**B**) Quantification of preserved axons in transverse sciatic nerve sections of distal nerve stumps from mice with the indicated genotypes. Each symbol in the scatter dot plots represents the quantification from one animal (% of control axon numbers in micrographs from uninjured contralateral nerve preparations for each animal). All data were obtained from experimental animals between 3 and 12 months of age.

The online version of this article includes the following source data and figure supplement(s) for figure 1:

**Source data 1.** Numerical source data (axon survival quantification) for graphs shown in *Figure 1B*.

**Figure supplement 1.** Normal rates of nerve injury-induced axon disintegration in DR6-deficient mice at earlier stages of Wallerian degeneration.

**Figure supplement 1—source data 1.** Numerical source data (relative axon survival quantification) for graphs shown in *Figure 1—figure supplement 1A and B*.

**Figure supplement 2.** Normal JNK activation in sciatic nerves from DR6-deficient mice.

**Figure supplement 2—source data 1.** Numerical source data (densitometric p-JNK/JNK ratio quantification) for graphs shown in *Figure 1—figure supplement 2B and D*.

**Figure supplement 2—source data 2.** TIF file with original western blots and boxes indicating the relevant bands shown in *Figure 1—figure*

*Figure 1 continued on next page*

*Figure 1 continued*

***supplement 2A and C.***

**Figure supplement 2—source data 3.** Original files for western blot analysis displayed in ***Figure 1—figure supplement 2A and C.***

in contrast to the *in vivo* data, no phenotypic penetrance effects in delaying AxD have been reported using *in vitro* assays in the previous DR6 knockout study focusing on WD (***Gamage et al., 2017***).

We confirmed the loss of *Tnfrsf21* mRNA and DR6 protein in dorsal root ganglia explants prepared from *Tnfrsf21*$^{\Delta Ex2-3/\Delta Ex2-3}$, and additionally *Tnfrsf21*$^{Ex2LoxP/Ex2LoxP}$ embryos infected with a lentivirus expressing Cre recombinase (dubbed DR6 $^{Cre}$) (***Figure 3A and D***, and not shown). However, time-lapse analysis of neurite disintegration after axotomy demonstrated no statistically significant differences between mutant and control preparations from the two models, with the only exception of slightly accelerated AxD in the DR6$^{Cre}$ preparations with the lentiviral Cre recombinase expression 6 hr after axotomy (***Figure 3B, C, E, and F***). This contrasts with the robust axonal protection observed in similar primary culture models devoid of Phr1/Mycbp2 or in cultures with expression of Wld$^S$ protein variants we previously reported (***Babetto et al., 2013***; ***Babetto et al., 2010***; ***Beirowski et al., 2009***). Together, these findings indicate that the loss of DR6 does not confer resistance to AxD in an established neuronal culture model.

## Discussion

The main motivation for this study was to determine if the glial injury responses previously suggested as abnormal in DR6-deficient mice play a role in the extended axon survival observed during WD (***Gamage et al., 2017***). Unexpectedly, despite using the same DR6 null mutant mouse line as the original study, we did not observe delayed AxD or the aberrant myelin remodeling described in that work. Our findings demonstrate that DR6 does not function as a pro-degenerative component within the axonal auto-destruction pathway, nor does it play a significant role in regulating the glial injury responses and myelin remodeling dynamics characteristic of WD in the peripheral nervous system (PNS). Our study underscores the importance of rigorous validation when assigning functional relevance to molecules with possible roles in the WD pathway.

The axon with its associated glial cells forms a specialized structural and functional unit that enables long-range neural communication. Much like how compromised cells initiate self-destruction, injured axons undergo a related but molecularly distinct process during WD, characterized by rapid metabolic collapse (***Yang et al., 2015***; ***Gerdts et al., 2015***). We recently showed that SCs modulate this process through early injury responses that enhance metabolic coupling with axons (***Babetto et al., 2020***). Importantly, the WD pathway of axonal self-destruction is believed to contribute significantly to the progression of various neurodegenerative diseases, highlighting both its cell-autonomous and non-cell-autonomous suppression as promising therapeutic strategies. To date, apart from expression of the aberrant fusion protein Wld$^S$, only four endogenous molecules – Nmnat2, SARM1, Phr1/Mycbp2, and DR6 – have been implicated to play a significant role in the WD pathway in mammals (***Gamage et al., 2017***; ***Babetto et al., 2013***; ***Osterloh et al., 2012***; ***Milde et al., 2013***). Their overexpression or inactivation has been reported to confer drastic resistance to experimentally induced WD, with disconnected axons in the PNS surviving for days to weeks, unlike wild-type axons, which fully degenerate within 48–72 hr on structural level. While Wld$^S$, Nmnat2, SARM1, and Phr1/Mycbp2 are cell-autonomously involved in the common WD pathway leading to axonal metabolic failure, DR6 has emerged as a more elusive candidate. Initial reports suggested that DR6, along with its putative ligand amyloid precursor protein, mediates developmental axon pruning (***Marik et al., 2013***; ***Nikolaev et al., 2009***; ***Kallop et al., 2014***). However, this interpretation has been complicated by a recent retraction (***Nikolaev et al., 2024***) and conflicting findings (***Olsen et al., 2014***). A later study proposed a striking new role for DR6 in WD, reporting that severed axons in DR6 knockout mice remained structurally preserved for up to 4 weeks post-injury even though separated from their parent neuronal cell bodies (***Gamage et al., 2017***). However, this phenotype was observed in fewer than half the animals, suggesting incomplete penetrance. The axonal phenotype, when present, was similar to that in Wld$^S$ mice, which also showed incomplete phenotypic penetrance in this study (***Gamage et al., 2017***). Moreover, it was found that the subset of DR6-deficient mice with axon protection displayed abnormal patterns of myelin remodeling during delayed WD, suggesting aberrant SC injury

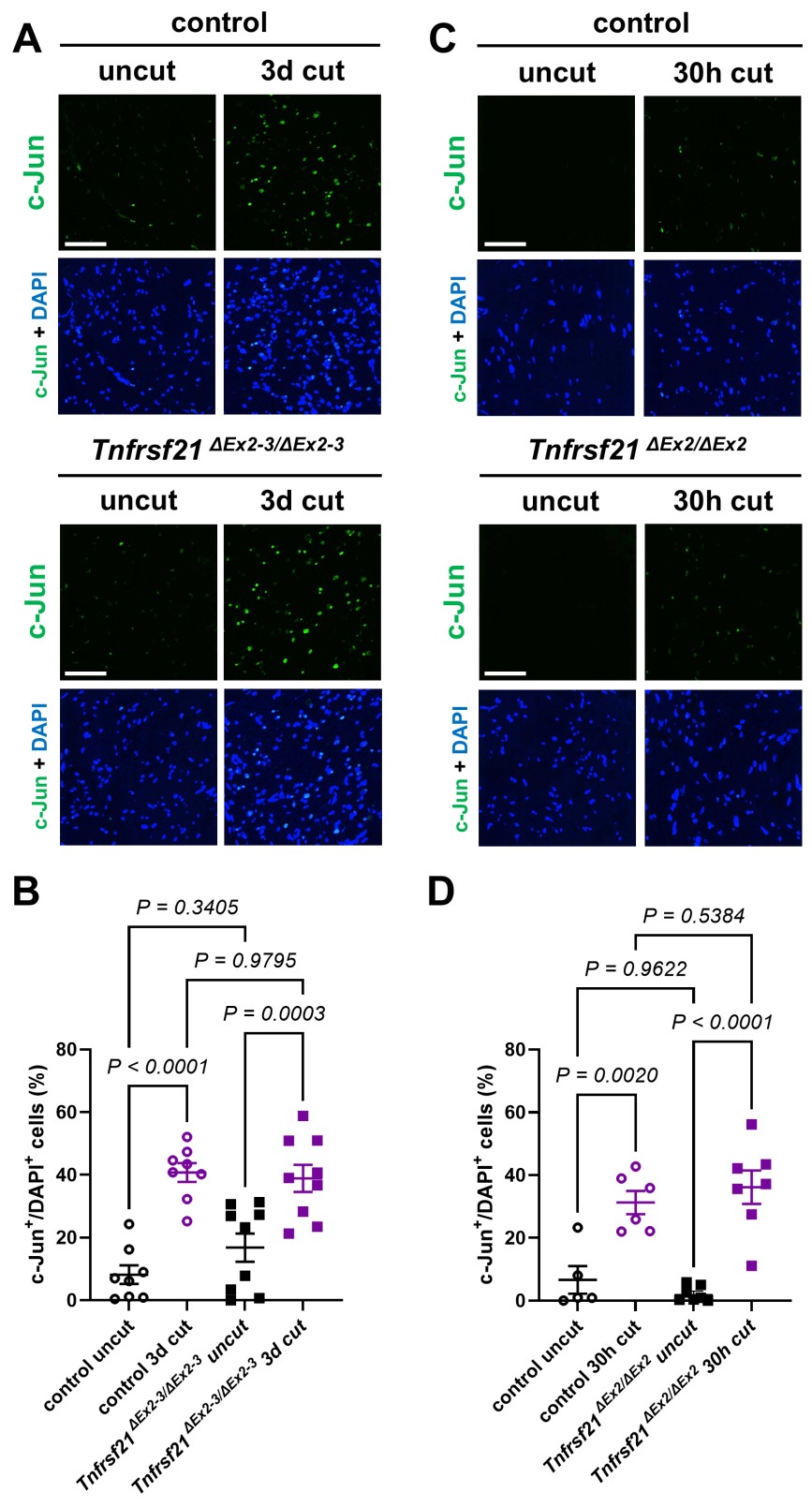

**Figure 2.** Normal Schwann cell c-Jun injury responses during Wallerian degeneration (WD) in mice lacking DR6. (**A, C**) Representative immunofluorescence confocal micrographs for c-Jun with DAPI nuclear counterstaining on transverse frozen sections from contralateral uninjured nerve (uncut) and distal sciatic nerve stumps 3 days and

*Figure 2 continued on next page*

*Figure 2 continued*

30 hr following axotomy in 3-months-old mice with the indicated genotypes. (**B, D**) Corresponding quantifications of percentage of c-Jun immunoreactive and DAPI⁺ cells on nerve sections. Scale bars: 50 µm.

The online version of this article includes the following source data and figure supplement(s) for figure 2:

**Source data 1.** Numerical source data (cell counts) for graphs shown in *Figure 2B and D*.

**Figure supplement 1.** Normal ultrastructure of transdifferentiated Schwann cells and normal myelin remodeling dynamics during Wallerian degeneration in DR6-deficient mice.

**Figure supplement 1—source data 1.** Numerical source data (myelin area quantification) for graphs shown in *Figure 2—figure supplement 1B and C*.

responses that could have contributed to the axonal phenotype. Furthermore, another study reported that neuronal DR6 expression regulates SC proliferation and inhibits developmental myelin formation in SCs (*Colombo et al., 2018*). These findings together raised the possibility that inactivation of DR6 might influence both neuronal and glial functions and their crosstalk during WD. However, to date, no subsequent studies have expanded upon or validated these discoveries.

In surprising contrast to the earlier data, our results provide compelling evidence that DR6 does not regulate AxD or glial injury responses during WD in the PNS. We employed a robust experimental design and have extensively characterized the kinetics of WD in two independent DR6 mouse knockout models using a high number of experimental animals and multiple *in vivo* and *in vitro* assays. For the first time, we present a comprehensive side-by-side comparison of key mouse mutants that have previously been shown to exhibit markedly delayed WD following nerve injury. We found no evidence that loss of DR6 expression, as confirmed by qRT-PCR and western blotting, antagonizes AxD during WD. Even under low-stringency conditions designed to detect subtle axonal protection, axon death progressed normally. These findings align with a prior electrophysiological study in the central nervous system, which suggested normal degeneration of optic nerve axons in DR6-deficient mice (*Fernandes et al., 2018*). We also did not find any changes in the SC injury responses and myelin reorganization known to be orchestrated by the transcription factor c-Jun during WD in the PNS (*Jessen and Mirsky, 2019*; *Arthur-Farraj et al., 2012*) in our DR6 mutant models at two post-injury time points that are frequently used to assess c-Jun expression and SC dedifferentiation characteristics. We therefore did not examine SC injury responses and myelin reorganization at 14 days after nerve injury, as performed in the previous study (*Gamage et al., 2017*), because no evidence of axonal protection was seen at earlier post-lesion stages.

The prominent discrepancy between our results and the prior study (*Gamage et al., 2017*) remains unresolved. One possibility is that a spontaneous germline mutation arose within the previously used mouse colony, conferring axonal protection in a subset of animals through a mechanism unrelated to DR6. Such mutation, perhaps conceptually similar to the one responsible for the Wld$^S$ phenotype that occurred spontaneously in the Olac breeding colony in England (*Lunn et al., 1989*), could then have propagated within the colony via autosomal dominant inheritance. Although we obtained our *Tnfrsf21$^{\Delta Ex2-3/\Delta Ex2-3}$* breeders directly from the laboratory that initially reported the protective phenotype (*Gamage et al., 2017*), it is conceivable that the specific mice we received did not carry the hypothetical second-site mutation. Alternatively, the phenotypic discrepancies may reflect the influences of unknown genetic modifiers, epigenetic factors, environmental conditions, or age-related differences that could affect the WD pathway. These variables could potentially explain the reported partial penetrance and the previously described developmental anomalies in DR6 null nerves (*Gamage et al., 2017*; *Colombo et al., 2018*). We also considered if changes in the nerve immune microenvironment might account for the discrepancies. For example, altered macrophage recruitment or activation, which is known to influence axon integrity, could contribute to differences in WD dynamics (*Vargas and Barres, 2007*). However, despite these possibilities, we note that we have never observed incomplete phenotypic penetrance effects in Wld$^S$ mice we developed (*Babetto et al., 2010*; *Beirowski et al., 2009*; *Mack et al., 2001*; *Beirowski et al., 2010*; *Conforti et al., 2009*) or in mutants lacking SARM1 or Phr1/Mycbp2 (*Babetto et al., 2013*), either in our hands or in reports from other laboratories over the last decades. This supports the view that the manipulation of these core components of an evolutionary conserved pathway produces fully penetrant protective phenotypes. Nonetheless,

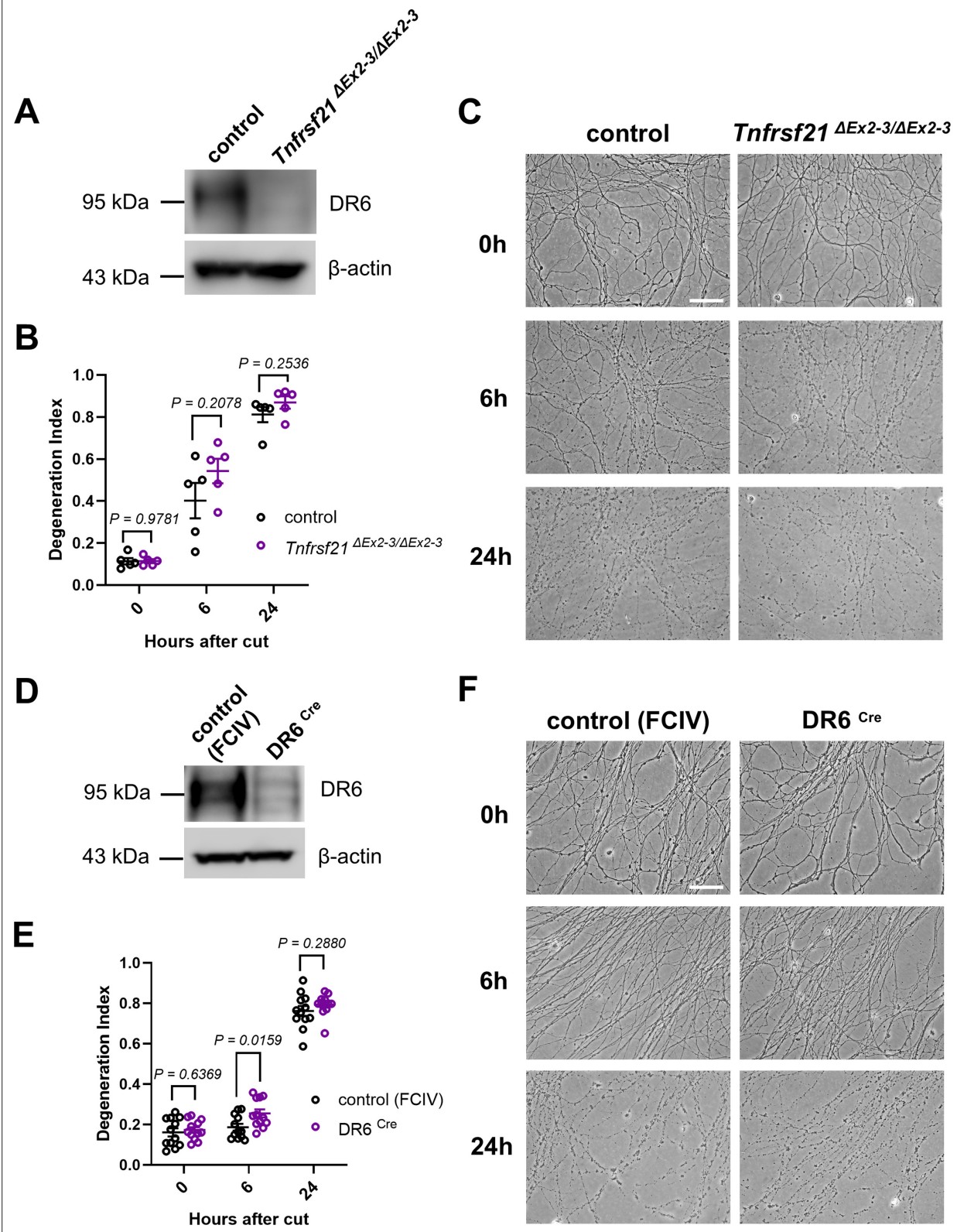

**Figure 3.** Normal degeneration kinetics of injured dorsal root ganglion (DRG) neurites in primary neuronal cultures from DR6 knockout mice. (**A, D**) Western blot analysis (cropped blot images) of DR6 protein expression in dorsal root ganglia (DRGs) from $Tnfrsf21^{\Delta Ex2-3/\Delta Ex2-3}$ embryos and from $Tnfrsf21^{Ex2LoxP/Ex2LoxP}$ embryos infected with a lentivirus expressing Cre recombinase (DR6$^{Cre}$), together with the respective control preparations. (**B, E**) Time course of neurite fragmentation, quantified as degeneration index (DI) (see Materials and methods) in DRG preparations from embryos with the

*Figure 3 continued on next page*

*Figure 3 continued*

indicated genotypes or lentiviral infection conditions. The data points shown in (**B**) represent the averaged neurite DI values calculated from multiple DRG preparations from each embryo at the indicated time point after injury (one data point = one embryo). The data points shown in (**E**) represent the averaged neurite DI values calculated from multiple micrographs acquired from each DRG preparation in a cell culture well at the indicated time point after injury (one data point = one well). (**C, F**) Representative phase-contrast micrographs of disconnected DRG neurites from embryos with the indicated genotypes and lentiviral infection conditions at the indicated time points after axotomy. Scale bars: 50 μm.

The online version of this article includes the following source data for figure 3:

**Source data 1.** Numerical source data (degeneration index) for graphs shown in *Figure 3B and E*.

**Source data 2.** TIF file with original western blots and boxes indicating the relevant bands shown in *Figure 3A and D*.

**Source data 3.** Original files for western blot analysis displayed in *Figure 3A and D*.

environmental modulation of axon protection in Wld$^S$ mice has been described, although the reported effects are relatively modest (*Perry et al., 1990*).

DR6 inhibition has shown therapeutic benefit in several models of neurodegeneration, including amyotrophic lateral sclerosis (*Huang et al., 2013*; *Mishra et al., 2020*), multiple sclerosis (*Mi et al., 2011*; *Schmidt et al., 2005*), prion disease (*Wang et al., 2015*), and *in vitro* amyloid-beta toxicity (*Hu et al., 2013*). However, no protective effects were observed in two separate mouse models of Alzheimer's disease (*Kallop et al., 2014*). While these effects have often been attributed to reduced neuronal death (*Hu et al., 2013*; *Mishra et al., 2020*), changes in immune function (*Schmidt et al., 2005*), or enhanced myelination (*Mi et al., 2011*), the suggestion that DR6 suppression stabilizes injured axons raised the possibility that axon protection itself may have contributed to disease amelioration. Our data argue against this interpretation in the context of WD. Instead, as mentioned earlier, DR6 may play a regulatory role in alternative forms of AxD, such as axon pruning, which is known to shape and refine neural circuits during development and may also be reactivated under pathological conditions (*Riccomagno and Kolodkin, 2015*; *Luo and O'Leary, 2005*). Future studies using disease-specific models will be necessary to explore this hypothesis.

## Materials and methods
### Mice

All experiments were performed in compliance with the Association for Assessment of Laboratory Animal Care policies and approved by The Ohio State University College of Medicine Animal Care and Use Committee. Mice were housed under specific pathogen-free conditions at 70°F, 50% room humidity, 12 hr light/12 hr dark cycle, and received *ad libitum* access to water and food. Mice with different genotypes were group-housed in separate cages. The mice used in this study did not undergo any procedures before their stated use. The mice were of mixed sexes. In previous studies, we found no significant differences in the rates of injury-induced AxD between female and male mice. Mice within individual comparative *in vivo* experiments were littermates or, if use of littermates was not possible because of litter sizes not large enough, mice from different litters were age-matched for comparative experiments. Controls were regarded as littermates or age-matched mice from different litters carrying wild-type alleles or floxed alleles with no Cre recombinase expression for the respective genotype groups. Constitutive DR6 knockout mice with deletion of exons 2 and 3 of *Tnfrsf21* (*Tnfrsf21$^{\Delta Ex2-3/\Delta Ex2-3}$*, on C57BL/6J.129S mixed background) and conditional DR6 mice with exon 2 flanked by loxP sites (*Tnfrsf21$^{Ex2LoxP/Ex2LoxP}$*) (genetic background unknown) were generated by Genentech (*Zhao et al., 2001*; *Tam et al., 2012*). To generate *Tnfrsf21$^{\Delta Ex2/\Delta Ex2}$* mice, we crossed *Tnfrsf21$^{Ex2LoxP/Ex2LoxP}$* mice to CMV-Cre transgenic mice (strain no. 006054, The Jackson Laboratory, made congenic on C57Bl/6J background). For this study, we additionally used heterozygous Wld$^S$ (*Mack et al., 2001*), Phr1 aKO (Phr1 adult conditional knockout) (*Babetto et al., 2013*), and *Sarm1$^{-/-}$* mice (strain no. 018069, The Jackson Laboratory, made congenic on C57Bl/6J background). Genotyping was performed by PCR strategies using standard procedures and appropriate primers (sequences available upon request).

### Multiplex TaqMan qRT-PCR

Brain tissue from *Tnfrsf21$^{\Delta Ex2-3/\Delta Ex2-3}$* and *Tnfrsf21$^{\Delta Ex2/\Delta Ex2}$* mutants and the respective control mice were homogenized using a Bullet Blender (Next Advance, model BBX24B) and RNAase-free zirconium

oxide beads (0.5 mm diameter, Next Advance). Total brain RNA was extracted using PureLink RNA Mini Kit (Thermo Fisher Scientific) according to the manufacturer's guidelines, and RNA concentration and purity were assessed using a NanoDrop One spectrophotometer (Thermo Fisher Scientific). To eliminate potential genomic DNA contamination, 4 µg of total RNA per sample was treated with EzDNAase (Invitrogen) according to the manufacturer's instructions. The resulting RNA was split into two aliquots: 2 µg were reverse-transcribed into cDNA using SuperScript VILO Master Mix (Thermo Fisher Scientific), and the remaining 2 µg underwent the same reaction without the reverse transcriptase enzyme to generate a no-RT control. Reaction times and volumes followed the manufacturer's guidelines. DR6 expression was tested with TaqMan gene expression predesigned assay Mmoo446361_m1 (Thermo Fisher Scientific) with PAM-MGB probe (*Hu et al., 2013*), and the housekeeping gene peptidyl-prolyl isomerase A (PPIA) was tested with TaqMan gene expression assay Mm02342430_g1 (Thermo Fisher Scientific) with VIC-MGP probe. Briefly, in each well of a 96-well plate (MicroAmp Optical 96-well reaction plate with barcode, Applied Biosystems), 1 µl of cDNA was tested with 9 µl of Mastermix containing 0.5 µl of TaqMan probe, using a QuantStudio 3 Real Time PCR instrument (Applied Biosystems). The thermal cycling conditions included an uracil-N glycosylase incubation step (50°C for 2 min), followed by polymerase activation (95°C for 20 s), and 40 cycles of PCR (denaturation at 95°C for 3 s and annealing/extension at 60°C for 30 s). ROX reference reading (passive reference) and cycle-threshold (CT) values were collected, visualized with QuantStudio 3 software, and analyzed with the ΔΔCT method. DR6 expression levels were calculated and normalized to PPIA levels in the same well of the plate.

## Western blot analysis

Mouse brains were quickly dissected and homogenized using a Polytron homogenizer (PT 1200C), followed by sonication with a Qsonica sonicator (Q500) equipped with a microtip. Sciatic nerves were quickly dissected, the epineurium removed in ice-cold phosphate-buffered saline (PBS), and directly sonicated as above. Both brains and nerves were homogenized in RIPA buffer supplemented with protease and phosphatase inhibitors. Embryonic dorsal root ganglia (eDRGs) preparations were washed with PBS to remove culture medium and then manually collected with forceps, placed in a tube with RIPA buffer supplemented with protease and phosphatase inhibitors, and then vortexed. After lysis, samples were spun at 10,000×*g* for 10 min at 4°C to remove insolubilized material. Western blotting of tissue lysates was performed using Bolt Mini gels and Mini Blot wet transfer modules (Invitrogen) according to the manufacturer's instructions. For detection of the DR6 protein, tissue and cell homogenates were processed under denaturing conditions on Tris-Acetate gels according to the manufacturer's protocol optimized for large molecular weight proteins (Thermo Fisher Scientific). For detection of JNK and p-JNK (Thr183/Tyr185) in nerve lysates, proteins were separated on 4–12% Bis-Tris gradient gels (Thermo Fisher Scientific). The following antibodies were used in a solution of 5% bovine serum albumin in Tris-buffered saline (TBS) with 0.1% of Tween20: anti-DR6 clone 6B6 (1:500, catalog no. MABC1594, Millipore Sigma), anti-phospho-SAPK/JNK (Thr183/Tyr185) clone 81E11 (1:1000, catalog no. 4668, Cell Signaling), anti-SAPK/JNK (1:1000, catalog no. 9252, Cell Signaling), anti-beta-actin (1:10,000, catalog no. A5316, Millipore Sigma). Horseradish peroxidase-conjugated secondary antibodies from Cell Signaling Technology and Jackson ImmunoResearch Laboratories were used for signal detection. Blot documentation was performed with a G:Box mini 6 digital imaging system (Syngene). The integrated band intensities of protein bands were determined using ImageJ software to calculate p-JNK(Thr183/Tyr185)/JNK ratios.

## Histomorphometry of intact nerves

The embedding of nerve samples in Araldite 502 epoxy resin (Polysciences) and subsequent cutting and staining of 500 nm semithin cross sections was performed as described previously (*Babetto et al., 2020*; *Beirowski et al., 2017*; *Beirowski et al., 2011*). High-resolution tile scans of the entire sciatic or tibial nerve cross sections were acquired with a DMi8 digital imaging system equipped with a 100× high-numerical aperture objective and oil immersion condenser using the Application Suite X 3.7.2 software (Leica Microsystems). The quantification of thinly myelinated axons on entire sciatic nerves from P1 mouse pups was performed manually using the ImageJ Cell Counter plug-in. The g-ratios of individual myelinated axons in sciatic or tibial nerves as a measure of myelin thickness were determined using a plug-in for the ImageJ software that allows semiautomated analysis of randomly selected

axons on nerve transverse sections (http://gratio.efil.de). One hundred randomly chosen fibers were measured per mouse nerve. Cumulative g-ratios were calculated for each mouse by averaging all individual g-ratios. Axon and fiber caliber distributions were quantified from sciatic or tibial nerve micrographs using ImageJ. The Wand tracing tool was used to manually delineate individual regions of interest (ROIs). For axon caliber measurements, the tool was applied to select the axonal area (excluding the myelin sheath), with the tolerance adjusted to accurately define each axon's boundary. For fiber caliber measurements, the entire fiber (axon plus myelin sheath) was selected using the same approach. The resulting ROI data were exported to Microsoft Excel, and calibers were calculated using the formula $A = \pi(d/2)^2$, where A is the area and d is the diameter. Binned distributions of axon and fiber calibers were calculated and visualized using GraphPad Prism software. All quantifications were conducted blinded to the genotypes of the animals.

## Electron microscopy

Transmission electron microscopy was carried out as previously described (*Babetto et al., 2020*; *Beirowski et al., 2017*). Electron micrographs of randomly selected areas on 85-μm-thick ultrathin nerve cross sections stained with uranyl acetate and lead citrate were taken with an FEI Tecnai transmission electron microscope.

## Unilateral sciatic nerve transection

Mice were deeply anesthetized using the SomnoSuite Low-Flow Anesthesia delivery system (Kent Scientific) operated with isoflurane. Right sciatic nerves were exposed and transected with surgical micro-scissors close to the sciatic notch, with the contralateral nerve serving as the control. The wound was closed with surgical thread or clips, and buprenorphine was administered as a postsurgery analgesic. Before nerve removal from humanely killed mice, the lesion site was inspected to verify complete transection. Distal sciatic and tibial nerve stumps and contralateral uninjured nerve segments were processed for semithin/electron microscopy or for immunofluorescence as described below.

## Analysis of axonal survival in axotomized nerve stumps

Axon survival after unilateral sciatic nerve transection was determined using highly established methods for the evaluation of axonal integrity (*Babetto et al., 2020*; *Beirowski et al., 2009*; *Mack et al., 2001*; *Beirowski et al., 2010*). Axon survival analysis 3 days post-axotomy was performed on semithin micrographs from randomly selected areas on sciatic nerve cross sections acquired with a DMi8 digital imaging system equipped with a 100× high-numerical aperture objective. One micrograph per mouse nerve was quantified. On each micrograph, the number of all structurally intact myelinated axons was manually counted using the Cell Counter plug-in of the ImageJ software. Axon survival was expressed as a percentage of intact axons relative to the total number of intact myelinated axons in corresponding uninjured control nerve micrographs. Axon survival at 30–36 hr after axotomy was assessed on montaged tile scan images of entire tibial nerves, acquired with a Leica DMi8 digital imaging system as described above, and similar to our previously published work (*Babetto et al., 2020*). Because WD at such early stages following axotomy is asynchronous, resulting in substantial heterogeneity of axons disintegration within a nerve (*Beirowski et al., 2004*), this method prevents the sampling bias that is associated with quantification of axons from individual micrographs taken from select areas of the nerve. Criteria for structurally intact axons after axotomy were uniform axoplasm with the presence of non-swollen mitochondria and normal myelin sheaths. Criteria for degenerated axons were degraded axoplasm, absence of mitochondria, and collapsed myelin sheaths. All structurally intact and all degenerated myelinated axons were counted on each tibial nerve montage, and their ratio was determined for each experimental animal (relative axon survival). Scoring was documented with the ImageJ software using the Cell Counter plug-in. The experimenter was blind to the genotypes of the mice during data acquisition. Relative axonal survival is reported as the percentage of the respective control group.

## Analysis of myelin remodeling in axotomized nerve stumps

Quantification of the area occupied by myelin (sheaths and myelin debris) in transverse semithin sections of axotomized sciatic and tibial nerves was performed using ImageJ. This analysis leverages the high contrast provided by osmium tetroxide and toluidine blue staining of lipid-rich myelin,

which enables reliable thresholding and binarization of myelin structures on nerve cross sections. The binarized images were used to calculate the proportion of the nerve cross-sectional area occupied by myelin. We used axotomized nerves from mice lacking c-Jun in SCs (c-Jun$^{loxP/loxP}$, P0$^{Cre}$) with suppressed SC transdifferentiation as a positive reference which shows abnormal myelin remodeling and clearance during WD. This results in significantly larger myelin areas occupied during WD as compared to control preparations.

## c-Jun immunofluorescence and quantification

Segments of control and injured sciatic nerves were immersion fixed in 4% paraformaldehyde/0.1 M PBS for 1 hr at 4°C, washed in PBS, cryoprotected in 30% sucrose, embedded in O.C.T. compound, and sectioned at 12 µm on a Leica cryostat (Leica Biosystems). The cross sections on adhesive glass slides were washed with TBS, permeabilized with 0.25% Triton X-100 and 0.5 M NH$_4$Cl in 0.05 M TBS for 10 min, blocked with 5% goat serum in 0.05 M TBS for 1 hr, and then incubated with anti-c-Jun antibody (1:200, catalog no. 9165, Cell Signaling Technology) diluted in 5% goat serum/0.05 M TBS overnight at 4°C. After washing in TBS, anti-rabbit secondary antibody coupled to Alexa Fluor 488 (1:500, catalog no. 115-545-003, Jackson ImmunoResearch Laboratories) was applied for 1 hr, the sections counterstained with DAPI (1:10,000, catalog no. 62247, Thermo Fisher Scientific) and mounted in VECTASHIELD Antifade Mounting Medium. Micrographs (Imaris image format) of the stained cross sections were captured with an Andor Dragonfly 200 spinning disc confocal imaging system (Oxford Instruments) and a 63× high-numerical aperture objective. Adjustments of brightness and contrast were applied with ImageJ and Microsoft PowerPoint equally across the entire image and were applied equally to control preparations for all the data presented. Two representative maximum-intensity z-projection images from randomly selected equal volumes were taken per nerve cross section and used for the subsequent quantification. The quantification of c-Jun$^+$/DAPI$^+$ cells versus c-Jun$^-$/DAPI$^+$ cells, used to calculate the percentage of cells showing c-Jun immunoreactivity, was carried out blind to the mouse genotypes.

## Preparation of mouse eDRG cultures

eDRGs were isolated from mouse E13.5 embryos. *Tnfrsf21*$^{wt/\Delta Ex2-3}$ females were timed-mated with *Tnfrsf21*$^{wt/\Delta Ex2-3}$ males, and embryos from each litter were processed individually and genotyped after eDRG plating. eDRGs from *Tnfrsf21*$^{wt/wt}$ embryos were used as control preparations. *Tnfrsf21*$^{Ex2LoxP/Ex2LoxP}$ females were time-mated with *Tnfrsf21*$^{Ex2LoxP/Ex2LoxP}$ males, and all extracted embryos from these litters were pooled together for eDRG isolation. eDRGs were dissociated and plated as previously described (*Shin and Cho, 2020*). A suspension of about 25,000 cells/µl was plated as a 2 µl droplet into a well within a 24-well plate. Neurons were cultured on poly-D-lysine and laminin-coated wells in Neurobasal medium supplemented with Glutamine (2 mM), Penicillin/Streptomycin, serum-free B27, nerve growth factor (50 ng/ml), and the anti-mitotic agents uridine and 5-Fluoro-2'-deoxyuridine (1 µM) to inhibit non-neuronal cell proliferation. At 2 days *in vitro* (DIV 2), neurons from *Tnfrsf21*$^{Ex2LoxP/Ex2LoxP}$ embryos were infected with either a control lentivirus (FCIV) or a lentivirus expressing Cre recombinase (CRE), both based on the same vector backbone and co-expressing EGFP, as previously described (*Babetto et al., 2013*). Infection efficiency was monitored via fluorescence microscopy based on EGFP expression.

## Analysis of *in vitro* AxD

At DIV 7, when eDRG neurites had developed an extended radial arborization pattern, neurites were transected from their neuronal cell bodies using a microblade under microscopic guidance. The region distal to the transection site was imaged by phase contrast at the indicated time points with a Leica DMi8 digital imaging system and processed as previously described (*Shin and Cho, 2020*). The micrographs were analyzed in ImageJ using a previously published degeneration index (DI) macro (*Sasaki et al., 2009*). This macro binarizes the images and calculates the area occupied by fragmented neurites, normalized to the total neurite area. The resulting DI value models the extent of structural axon disintegration during WD, with values above 0.8 reflecting complete axonal disintegration. For the comparison of *Tnfrsf21*$^{wt/wt}$ and *Tnfrsf21*$^{\Delta Ex2-3/\Delta Ex2-3}$ embryos, the time course of neurite fragmentation for each embryo was obtained by averaging the DI values calculated from multiple phase contrast micrographs taken from each cell culture well at each time point. For the comparison

of lentiviral-infected neurons from *Tnfrsf21*$^{Ex2LoxP/Ex2LoxP}$ embryos, the time course of neurite fragmentation in each eDRG cell culture well preparation was obtained by averaging the DI values calculated from multiple phase contrast micrographs taken from each cell culture well.

## Statistics and reproducibility

All shown micrographs (semithin and electron microscopy, immunofluorescence, phase contrast microscopy) are representative of at least three biological replicates (mice or cell culture preparations). All statistical analyses and data visualization were performed using GraphPad Prism software. An unpaired, two-tailed Student's t-test was used to compare two groups. A two-way analysis of variance (ANOVA) with Sidak's multiple comparisons test was used to compare more than two groups. Data are presented as arithmetic mean ± standard error of the mean (SEM). Statistical differences were considered to be significant when $p < 0.05$.

## Acknowledgements

We thank Genentech and Christopher Deppmann (University of Virginia) for providing the constitutive and conditional DR6 mutant mice, Konstantinos Tsesmelis for technical experimental assistance, Ekaterina Stepanova and Alessio Colombo for genotyping information, Devin Donich for help with the g-ratio measurements, and Sara Bombardelli and the University Laboratory and Animal Resources (ULAR) at The Ohio State University College of Medicine for mouse husbandry assistance. This study was supported by the NIH-NINDS grant R01NS123450 (to EB and BB).

## Additional information

### Funding

| Funder | Grant reference number | Author |
| --- | --- | --- |
| National Institute of Neurological Disorders and Stroke | R01NS123450 | Bogdan Beirowski Haoran Huang Elisabetta Babetto |

The funders had no role in study design, data collection and interpretation, or the decision to submit the work for publication.

### Author contributions

Bogdan Beirowski, Conceptualization, Resources, Data curation, Formal analysis, Supervision, Funding acquisition, Validation, Investigation, Visualization, Methodology, Writing – original draft, Project administration, Writing – review and editing, conceived and supervised the study, designed and planned the experiments, performed the experiments and analyzed and interpreted the data shown in the figures; Haoran Huang, Investigation, Visualization, performed the experiment for Fig. 2; Elisabetta Babetto, Conceptualization, Resources, Data curation, Formal analysis, Supervision, Funding acquisition, Validation, Investigation, Visualization, Methodology, Writing – original draft, Project administration, Writing – review and editing, conceived and supervised the study, designed and planned the experiments, performed the experiments and analyzed and interpreted the data shown in the figures

### Author ORCIDs

Bogdan Beirowski ⓘ https://orcid.org/0000-0002-1241-1777
Haoran Huang ⓘ https://orcid.org/0000-0001-6125-5996
Elisabetta Babetto ⓘ https://orcid.org/0000-0002-6385-8288

### Ethics

This study was performed in strict accordance with the recommendations in the Guide for the Care and Use of Laboratory Animals of the National Institutes of Health. All of the animals were handled according to approved institutional animal care and use committee (IACUC) protocol #2022A00000096 of The Ohio State University.

Reviewer #1 (Public review): https://doi.org/10.7554/eLife.108389.3.sa1
Reviewer #3 (Public review): https://doi.org/10.7554/eLife.108389.3.sa2
Author response https://doi.org/10.7554/eLife.108389.3.sa3

# Additional files

### Supplementary files
MDAR checklist

### Data availability
Numerical source data and raw western blot data have been provided for all figures in this article. The electron microscopy, immunofluorescence, and phase contrast raw imaging datasets have been deposited at Dryad (https://doi.org/10.5061/dryad.g79cnp652) and are publicly available.

The following dataset was generated:

| Author(s) | Year | Dataset title | Dataset URL | Database and Identifier |
|---|---|---|---|---|
| Beirowski B, Huang H, Babette E | 2026 | Data from: Death receptor 6 does not regulate axon degeneration and Schwann cell injury responses during Wallerian degeneration | https://doi.org/10.5061/dryad.g79cnp652 | Dryad Digital Repository, 10.5061/dryad.g79cnp652 |

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

## Appendix 1

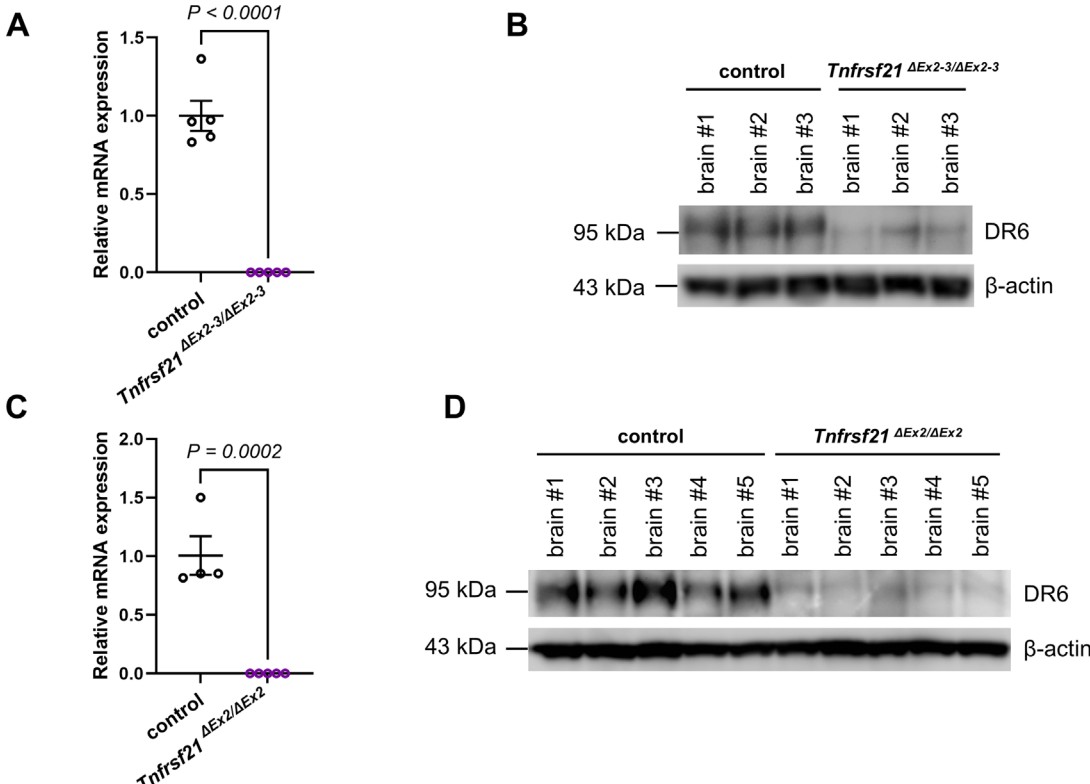

**Appendix 1—figure 1.** Disrupted DR6 expression in two DR6 knockout mouse models. (**A, C**) Quantitative real-time PCR analysis of relative brain *Tnfrsf21* mRNA levels from control and mutant mice with the indicated genotypes. Each dot represents the measurement from one mouse. (**B, D**) Western blot analysis (cropped blot images) of brain lysates from control and mutant mice with the indicated genotypes probed with the shown antibodies. Each western blot lane represents the brain lysate data from one individual mouse.

The online version of this article includes the following source data for appendix 1—figure 1:

**Appendix 1—figure 1—source data 1.** Numerical source data (relative mRNA expression) for graphs shown in *Appendix 1—figure 1A and C*.

**Appendix 1—figure 1—source data 2.** TIF file with original western blots and boxes indicating the relevant bands shown in *Appendix 1—figure 1B and D*.

**Appendix 1—figure 1—source data 3.** Original files for western blot analysis displayed in *Appendix 1—figure 1B and D*.

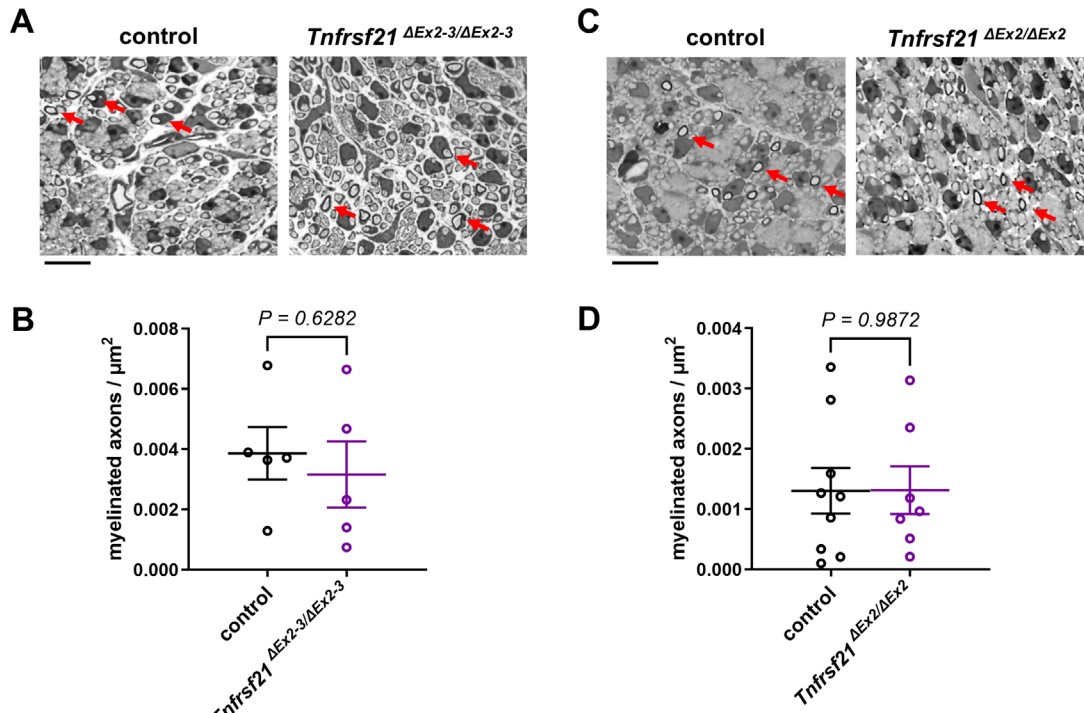

**Appendix 1—figure 2.** Normal developmental myelination in sciatic nerves from mice lacking DR6 at postnatal day 1. (**A, C**) Representative semithin micrographs of transverse sciatic nerve sections from control and mutant mouse pups at postnatal day 1 with the indicated genotypes. Red arrows depict examples of axons with nascent myelination. Scale bars: 10 μm. (**B, D**) Quantification of myelinated axons in transverse sciatic nerve sections from control and mutant pups at postnatal day 1 with the indicated genotypes. Each dot represents the quantification obtained from one mouse pup.

The online version of this article includes the following source data for appendix 1—figure 2:

**Appendix 1—figure 2—source data 1.** Numerical source data (myelinated axon counts) for graphs shown in *Appendix 1—figure 2B and D*.

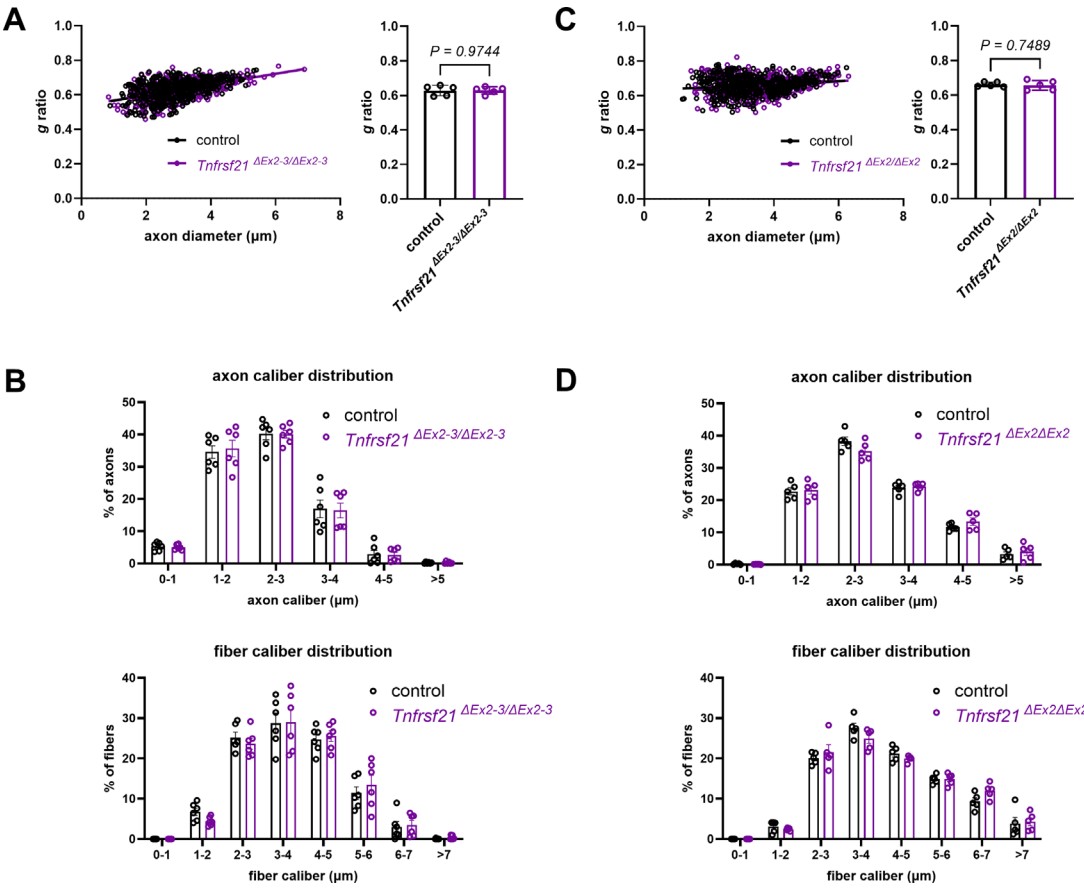

**Appendix 1—figure 3.** Normal nerve histomorphometry in DR6-deficient mice. (**A, C**) Quantification of g-ratios (left: scatter plots show g-ratios of individual myelinated axons as a function of axon diameter, right: corresponding cumulative g-ratios per animal) in tibial (**A**) and sciatic nerves (**C**) from 3-month-old control and mutant mice with the indicated genotypes. (**B, D**) Axon caliber and fiber caliber (axon plus myelin sheath) distributions in tibial (**B**) and sciatic nerves (**D**) from 3-month-old control and mutant mice with the indicated genotypes. Each dot represents the measurement from one mouse.

The online version of this article includes the following source data for appendix 1—figure 3:

**Appendix 1—figure 3—source data 1.** Numerical source data (histomorphometry quantification) for graphs shown in *Appendix 1—figure 3*.

