## [Editor Report · eLife Assessment]

In this **valuable** study, through carefully executed and rigorously controlled experiments, the authors challenged a previously reported role of the Death Receptor 6 (DR6/Tnfrsf21) in Wallerian degeneration (WD). Using two DR6 knockout mouse lines and multiple WD assays, both *in vitro* and *in vivo*, the authors provided **convincing** evidence that loss of DR6 in mice does not protect peripheral axons from WD after injury, at least in the specific contexts of the mice and analyses performed in this study. Due to the lack of certain specific parameters from previous studies (sex, age, mouse strains etc.), the exact reasons underlying the observed inconsistencies between current and previous reports on the protective effects of DR6 remains to be determined. Overall, this is a carefully executed study providing invaluable information toward understanding DR6's role (or lack thereof) in axon degeneration.

---

## [Referee Report · Reviewer #1 (Public review)]

Summary:

The authors show that genetic deletion of the orphan tumor necrosis factor receptor DR6 in mice does not protect peripheral axons against degeneration after axotomy. Similarly, Schwann cells in DR6 mutant mice react to axotomy similarly to wild type controls. These negative results are important because previous work has indicated that loss or inhibition of DR6 is protective in disease models and also against Wallerian degeneration of axons following injury. This carefully executed counterexample is important for the field to consider.

Strengths:

A strength of the paper is the use of two independent mouse strains that knockout DR6 in slightly different ways. The authors confirm that DR6 mRNA is absent in these models (western blots for DR6 protein are less convincingly null, but given the absence of mRNA, this is likely an issue of antibody specificity). One of the DR6 knockout strains used is the same strain used in a previous paper examining the effects of DR6 on Wallerian degeneration.

The authors use a series of established assays to evaluate axon degeneration, including light and electron microscopy on nerve histological samples and cultured dorsal root ganglion neurons in which axons are mechanically severed and degeneration is scored in time lapse microscopy. These assays consistently show a lack of effect of loss of DR6 on Wallerian degeneration in both mouse strains examined.

Additional strengths are that the authors examine both the axonal response and the Schwann cell response to axotomy and use both in vivo and in vitro assays.

Therefore, these experiments, the author's data support their conclusion that loss of DR6 does not protect against Wallerian degeneration.

Weaknesses:

A weakness of this paper is that no effort is made to determine why the results presented here may differ from previous studies. A notable possibility is that the original mouse strain that showed 5 of 13 mice being protected from Wallerian degeneration was studies on a segregating C57BL/6.129S background.

Finally, it is important to note that previously reported effects of DR6 inhibition, such as protection of cultured cortical neurons from beta-amyloid toxicity, are not necessarily the same as Wallerian degeneration of axons distal to an injury studied here. The negative results presented here showing that loss of DR6 is not protective against Wallerian degeneration induced by injury are important given the interest in DR6 as a therapeutic target. However, care should be taken in attempting to extrapolate these results to other disease contexts such as ALS or Alzheimer's disease.

---

## [Referee Report · Reviewer #3 (Public review)]

Summary:

The authors revisit the role of DR6 in axon degeneration following physical injury (Wallerian degeneration), examining both its effects on axons and its role in regulating the Schwann cell response to injury. Surprisingly, and in contrast to previous studies, they find that DR6 deletion does not delay the rate of axon degeneration after injury, suggesting that DR6 is not a mediator of this process.

Overall, this is a valuable study. As the authors note, the current literature on DR6 is inconsistent, and these results provide useful new data and clarification. This work will help other researchers interpret their own data and re-evaluate studies related to DR6 and axon degeneration.

Strengths:

(1) The use of two independent DR6 knockout mouse models strengthens the conclusions, particularly when reporting the absence of a phenotype.

(2) The focus on early time points after injury addresses a key limitation of previous studies. This approach reduces the risk of missing subtle protective phenotypes and avoids confounding results with regenerating axons at later time points after axotomy.

Comments on revisions:

I thank the authors for their thorough responses to my previous comments. The revisions have addressed the points raised and have improved the clarity and overall quality of the manuscript. I appreciate the effort taken to strengthen the presentation of the work.

---

## [Author Response]

The following is the authors’ response to the original reviews.

**Public Reviews:**

**Reviewer #1 (Public review):**
Summary:The authors show that genetic deletion of the orphan tumor necrosis factor receptor DR6 in mice does not protect peripheral axons against degeneration after axotomy. Similarly, Schwann cells in DR6 mutant mice react to axotomy similarly to wild-type controls. These negative results are important because previous work has indicated that loss or inhibition of DR6 is protective in disease models and also against Wallerian degeneration of axons following injury. This carefully executed counterexample is important for the field to consider.Strengths:A strength of the paper is the use of two independent mouse strains that knock out DR6 in slightly different ways. The authors confirm that DR6 mRNA is absent in these models (western blots for DR6 protein are less convincingly null, but given the absence of mRNA, this is likely an issue of antibody specificity). One of the DR6 knockout strains used is the same strain used in a previous paper examining the effects of DR6 on Wallerian degeneration.The authors use a series of established assays to evaluate axon degeneration, including light and electron microscopy on nerve histological samples and cultured dorsal root ganglion neurons in which axons are mechanically severed and degeneration is scored in time-lapse microscopy. These assays consistently show a lack of effect of loss of DR6 on Wallerian degeneration in both mouse strains examined.Therefore, in the specific context of these experiments, the author's data support their conclusion that loss of DR6 does not protect against Wallerian degeneration.Weaknesses:(1) The major weaknesses of this paper include the tone of correcting previously erroneous results and the lack of reporting on important details around animal experiments that would help determine whether the results here really are discordant with previous studies, and if so, why.The authors do not report the genetic strain background of the mice used, the sex distributions of their experimental cohorts, or the age of the mice at the time the experiments were performed. All of these are important variables.

(Response 1) We thank the reviewer for emphasizing the importance of reporting the sex, age, and genetic background of the experimental animals used in our axon protection analyses. We have incorporated this information into the revised manuscript wherever available. The sole exception concerns the genetic background of the conditional DR6 mice generated by Genentech, which remains unknown. The original publication describing these mice (Tam et al., 2012, Dev Cell, PMID 22340501) did not report this information, and we were unable to obtain it directly from Genentech. Details regarding the genetic background of the Wld^S^ and aPhr1 mutant mice are provided in their respective original publications, which are cited in our manuscript. Because the Gamage et al. study from the Deppmann laboratory did not report the sex or age of the animals used, we were unable to assess whether these variables might contribute to the differences observed between the two studies. Moreover, we are not aware of published evidence identifying sex or age as modifiers of structural axon preservation in axotomized peripheral nerve stumps in mouse models of delayed Wallerian degeneration. Furthermore, in the original publications describing the phenotypes of transgenic Nmnat2 and Wld^S^ mice, as well as Sarm1 or Phr1 knockout mice, sex and age of the animals used in the Wallerian degeneration assays were not reported (PMIDs 23995269, 12106171, 22678360, 23665224). Although, to our knowledge, no large-scale systematic studies have been conducted, over the last 15 years we have never observed any sex-based differences in Wallerian degeneration phenotypes in these mutants exhibiting prominent axon protection. This topic was discussed informally at conferences, and we are also not aware of other investigators having observed such effects.

In response to the reviewer’s comment regarding “tone”, we made sure that our data and interpretations are presented in a professional, balanced, and objective manner, including a detailed discussion of potential alternative explanations for the discrepant findings.

(2) The DR6 knockout strain reported in Gamage et al. (2017) was on a C57BL/6.129S segregating background. Gamage et al. reported that loss of DR6 protected axons from Wallerian degeneration for up to 4 weeks, but importantly, only in 38.5% (5 out of 13) mice they examined. In the present paper, the authors speculate on possible causes for differences between the lack of effect seen here and the effects reported in Gamage et al., including possible spontaneous background mutations, epigenetic changes, genetic modifiers, neuroinflammation, and environmental differences. A likely explanation of the incomplete penetrance reported by Gamage et al. is the segregating genetic background and the presence of modifier loci between C57BL/6 and 129S. The authors do not report the genetic background of the mice used in this study, other than to note that the knockout strain was provided by the group in Gamage et al. However, if, for example, that mutation has been made congenic on C57BL/6 in the intervening years, this would be important to know. One could also argue that the results presented here are consistent with 8 out of 13 mice presented in Gamage et al.

(Response 2) As noted above, we now provide information on the genetic background of the mice in the revised manuscript, where available. We have not backcrossed the constitutive DR6 knockout mice obtained from the Deppmann laboratory (Gamage et al.) to a C57BL/6 background; our colony was maintained primarily through intercrosses of heterozygous animals. Similarly, the conditional DR6 mutant mice used in this study were also not backcrossed to C57BL/6 mice.

We respectfully hold a different view regarding the reviewer’s final point. We understand it is not appropriate to infer consistency between two datasets by disregarding the subset of results that do not align. By the same logic, it would be flawed to draw conclusions from the Gamage et al. study based solely on the single Wld^S^ mouse out of five that did not show axon preservation after nerve injury. Selectively omitting conflicting data does not provide a valid basis for establishing phenotype concordance across studies.

To further strengthen our study, we note that we performed additional analyses on three more nerve samples from constitutive DR6 null mice during the revision process and have incorporated the resulting data in Fig. 1.

(3) Age is also an important variable. The protective effects of the spontaneous WldS mutation decrease with age, for example. It is unclear whether the possible protective effects of DR6 also change with age; perhaps this could explain the variable response seen in Gamage et al. and the lack of response seen here.

(Response 3) As discussed above, we now provide the age information for the mice used for the Wallerian degeneration assessments in the respective figure legends. To our knowledge, there are no prior reports suggesting that age is a significant determinant of structural axon preservation in the indicated mutants. Electrophysiological function and neuromuscular junction preservation decrease with age in axotomized Wld^S^ mice (e.g., PMIDs 12231635, 19158292, 15654865), but these parameters are not subject of our study, and we have not studied them. Unfortunately, a direct comparison of ages between our DR6 mutant mice and those used in Gamage et al. (2017) is not possible, as the earlier study from the Deppmann laboratory did not report this information.

(4) It is unclear if sex is a factor, but this is part of why it should be reported.

(Response 4) We now report the requested sex information for our axon preservation analyses during nerve injury-induced Wallerian degeneration in the DR6 mouse models in Figs. 1 and 2.

(5) The authors also state that they do not see differences in the Schwann cell response to injury in the absence of DR6 that were reported in Gamage et al., but this is not an accurate comparison. In Gamage et al., they examined Schwann cells around axons that were protected from degeneration 2 and 4 weeks post-injury. Those axons had much thinner myelin, in contrast to axons protected by WldS or loss of Sarm1, where the myelin thickness remained relatively normal. Thus, Gamage et al. concluded that the protection of axons from degeneration and the preservation of Schwann cell myelin thickness are separate processes. Here, since no axon protection was seen, the same analysis cannot be done, and we can only say that when axons degenerate, the Schwann cells respond the same whether DR6 is expressed or not.

(Response 5) We appreciate the reviewer’s detailed comments. Our intention was not to directly compare our findings with those of Gamage et al. regarding the myelin behavior at these time points (because we never observed axon protection), but rather to note that we did not observe any DR6-dependent alterations in Schwann cell responses under conditions where axons undergo normal Wallerian degeneration. As the reviewer correctly points out, Gamage et al. analyzed Schwann cell myelin surrounding axons that were protected from degeneration for extended periods, a context fundamentally different from the complete lack of axon protection observed in our DR6-deficient models. Therefore, the specific dissociation between axon preservation and myelin maintenance claimed by Gamage et al. cannot be evaluated in our study. A statement to make this point clearer has been incorporated in the revised manuscript.

We fully agree with the reviewer’s concluding point: in our experiments, once axons degenerate, Schwann cell responses proceed similarly regardless of DR6 expression. This agreement reinforces one of the central conclusions of our work.

(6) The authors also take issue with Colombo et al. (2018), where it was reported that there is an increase in axon diameter and a change in the g-ratio (axon diameter to fiber diameter - the axon + myelin) in peripheral nerves in DR6 knockout mice. This change resulted in a small population of abnormally large axons that had thinner myelin than one would expect for their size. The change in g-ratio was specific to these axons and driven by the increased axon diameter, not decreased myelin thickness, although those two factors are normally loosely correlated. Here, the authors report no changes in axon size or g-ratio, but this could also be due to how the distribution of axon sizes was binned for analysis, and looking at individual data points in supplemental figure 3A, there are axons in the DR6 knockout mice that are larger than any axons in wild type. Thus, this discrepancy may be down to specifics and how statistics were performed or how histograms were binned, but it is unclear if the results presented here are dramatically at odds with the results in Colombo et al. (2018).

(Response 6) Several points raised by the reviewer appear to reflect differences in interpretation of the findings reported in Colombo et al. (2018). That study did not report altered myelination in DR6 null mice at stages when myelination is largely complete (P21). Instead, modest changes were observed at P1, which were reduced by P7, and P21 mutants were reported to be indistinguishable from controls. No analyses of peripheral nerves in older animals were presented, and the authors concluded in the discussion that myelination in young adult DR6 null mice appears normal. In contrast, our analysis of constitutive DR6 null mice at P1 does not reproduce the increase in the number of myelinated fibers per unit area reported by Colombo et al. We obtained similar results in the independent conditional DR6 knockout mouse line. Differences in nerve tissue processing, embedding, staining, or in the microscopic imaging and quantification of thinly myelinated axons in P1 sciatic nerve cross-sections may have contributed to the observed discrepancy. However, because the relevant methodological details were not described in Colombo et al., the underlying reasons for these differences cannot be determined and remain speculative.

(7) Finally, it is important to note that previously reported effects of DR6 inhibition, such as protection of cultured cortical neurons from beta-amyloid toxicity, are not necessarily the same as Wallerian degeneration of axons distal to an injury studied here. The negative results presented here, showing that loss of DR6 is not protective against Wallerian degeneration induced by injury, are important given the interest in DR6 as a therapeutic target, but they are specific to these mice and this mechanism of induced axon degeneration. The extent to which these findings contradict previous work is difficult to assess due to the lack of detail in describing the mouse experiments, and care should be taken in attempting to extrapolate these results to other disease contexts, such as ALS or Alzheimer's disease.

(Response 7) We agree with the reviewer’s point and emphasize that our manuscript carefully differentiates our data regarding the function of DR6 in Wallerian degeneration from the potential involvement of DR6 in other forms of axon degeneration. Our findings do not conflict with previous work on DR6 in the context of *in vitro* beta-amyloid and prion toxicity as well as *in vitro* models of ALS and multiple sclerosis. We believe these distinctions are explicitly and appropriately articulated throughout the entire manuscript and in more detail in the discussion section.

**Reviewer #1 (Recommendations for the authors):**
(1) The authors should include additional information about the mice used, including strain background for both the DR6 mice and the Cre transgenes crossed into the DR6 conditional knockout, the age of the mice when the nerve crush experiments were performed, and the sex distributions of the experimental cohorts. This information is critical for reproducibility in animal experiments, and that point is compounded here, where the major focus of this paper is taking issue with the reproducibility of previous work.

(Response 8) This information has been included in the revision. See above responses.

(2) In the abstract, reference 5 is cited as a study on the response to Schwann cells to injury in a DR6 background, but this probably should be reference 10.

(Response 9) This typo has been corrected.

(3) "Site-by-site comparison" in line 201 should be side-by-side?

(Response 10) This typo has been corrected.

(4) The paper contains a lot of self-evaluative wording, "surprising contrast," "compelling evidence," "robust results." Whether those adjectives apply should be for the reader to decide, and a drier, more objective tone in the presentation would improve the paper.

(Response 11) We agree that excessive self-evaluative wording can weaken objectivity. In the manuscript, such phrasing is used sparingly and intentionally to highlight differences from previously published studies, guide the reader, and convey scholarly judgment. We do not consider this limited use to be counterproductive. The adjectives “surprising,” “compelling,” and “robust” each appear only one to three times across the entire manuscript, and the specific phrase “robust results” does not appear at all.

(5) In Figure 2A, DR6-/-, there is no significant difference, but there is also a lot of variability, and one could argue the authors are seeing axon protection comparable to WldS in 40% of their samples (2/5), which is very similar to Gamage et al.

(Response 12) We respectfully disagree with this reasoning as it relies on selectively emphasizing only a subset of the data. Please also see our response #2 for more detailed discussion.

(6) Overall, the data presented here are convincing and support the conclusions drawn, but the paper needs to focus more on the negative results at hand and less on bashing previous studies, particularly when the results presented here do definitively show that the previous studies were incorrect and plausible explanations for differences in outcome exist.

(Response 13) We have carefully revisited the wording of the manuscript and are confident that our emphasis remains on the central negative finding that DR6 does not regulate axon degeneration and Schwann cell injury responses during Wallerian degeneration. We do not believe the manuscript “bashes” previous studies; nonetheless, we thoroughly re-examined all relevant sections to ensure that our language is neutral, accurate, and non-inflammatory. We believe the current phrasing presents our interpretations in an appropriately balanced, objective, and professional manner.

**Reviewer #2 (Public review):**
Summary:This manuscript by Beirowski, Huang, and Babetto revisits the proposed role of Death Receptor 6 (DR6/Tnfrsf21) in Wallerian degeneration (WD). A prior study (Gamage et al., 2017) suggested that DR6 deletion delays axon degeneration and alters Schwann cell responses following peripheral nerve injury. Here, the authors comprehensively test this claim using two DR6 knockout mouse models (the line used in the earlier report plus a CMV-Cre derived floxed ko line) and multiple WD assays in vivo and in vitro, aligned with three positive controls, Sarm1 WldS and Phr1/Mycbp2 mutants. Contrary to the prior findings, they find no evidence that DR6 deletion affects axon degeneration kinetics or Schwann cell dynamics (assessed by cJun expression or [intact+degenerating] myelin abundance after injury) during WD. Importantly, in DRG explant assays, neurites from DR6-deficient mice degenerated at rates indistinguishable from controls. The authors conclude that DR6 is dispensable for WD, and that previously reported protective effects may have been due to confounding factors such as genetic background or spontaneous mutations.Strengths:The authors employ two independently generated DR6 knockout models, one overlapping with the previously published study, and confirm loss of DR6 expression by qPCR and Western blotting. Multiple complementary readouts of WD are applied (structural, ultrastructural, molecular, and functional), providing a robust test of the hypothesis.Comparisons are drawn with established positive controls (WldS, SARM1, Phr1/Mycbp2 mutants), reinforcing the validity of the assays.By directly addressing an influential but inconsistent prior report, the manuscript clarifies the role of DR6 and prevents potential misdirection of therapeutic strategies aimed at modulating WD in the PNS. The discussion thoughtfully considers possible explanations for the earlier results, including colony-specific second-site mutations that could explain the incomplete penetrance of the earlier reported phenotype of only 36%.Weaknesses:(1) The study focuses on peripheral nerves. The manuscript frequently refers to CNS studies to argue for consistency with their findings. It would be more accurate to frame PNS/CNS similarities as reminiscences rather than as consistencies (e.g., line 205ff in the Discussion).

(Response 14) Axon protection in all key genetic models of delayed axon degeneration, including Wld^S^, SARM1, Phr1/Mycbp2 mutants, has been demonstrated in both the peripheral and central nervous systems. This observation supports the view that core molecular mechanisms regulating axon degeneration are conserved across neuronal populations throughout the entire nervous system. We have scrutinized the wording in our manuscript and are not aware that we frequently refer to CNS studies in regards to axon degeneration. Nevertheless, we have replaced the term “consistent” to avoid potential ambiguity when we discuss the earlier study showing normal Wallerian degeneration in the optic nerves from DR6 knockout mice.

(2) The DRG explant assays are convincing, though the slight acceleration of degeneration in the DR6 floxed/Cre condition is intriguing (Figure 4E). Could the authors clarify whether this is statistically robust or biologically meaningful?

(Response 15) We thank the reviewer for noting this aspect of our *in vitro* data in Fig. 4. The difference observed in the DR6 floxed/Cre condition is statistically significant at the 6h time point following disconnection, as indicated by the p value shown in Fig. 4E. However, a similarly statistically significant acceleration of axon degeneration was not observed in DRG axotomy experiments using constitutive DR6 knockout preparations, although a trend toward more rapid axon breakdown is apparent at 6 h post-axotomy (Fig. 4B). These observations may suggest reduced stability of DR6-deficient axons in this specific neuron-only *in vitro* context. Further investigation would be required to determine the biological significance of this effect. In contrast, our *in vitro* quantitative analyses of the initiation and early phases of Wallerian degeneration (Fig. 2) revealed no evidence of accelerated axon disintegration in the DR6 mutant mouse models, highlighting potential differences between *in vitro* and *in vitro* systems.

(3) In the summary (line 43), the authors refer to Hu et al. (2013) (reference 5) as the study that previously reported AxD delay and SC response alteration after injury. However, this study did not investigate the PNS, and I believe the authors intended to reference Gamage et al. (2017) (reference 10) at this point.

(Response 16) Thanks for pointing this out. We have corrected this typo in the revised manuscript.

(4) In line 74ff of the results section, the authors claim that developmental myelination is not altered in DR6 mutants at postnatal day 1. However, the variability in Figure S2 appears substantial, and the group size seems underpowered to support this claim. Colombo et al. (2018) (reference 11) reported accelerated myelination at P1, but this study likewise appears underpowered. Possible reasons for these discrepancies and the large variability could be that only a defined cross-sectional area was quantified, rather than the entire nerve cross-section.

(Response 17) We confirm that the quantification of thinly myelinated axons was performed on entire sciatic nerves from P1 mouse pups, as described in the methods section in our original manuscript. The data shown in Fig. S2 were obtained from 5-9 pups per experimental group. Sample sizes were determined based on a priori power analyses using pilot data, which indicated that a minimum of five biological replicates was sufficient to detect statistically significant differences with acceptable confidence. Comparable sample sizes have been used in our previous studies and by other groups to assess early postnatal myelination (e.g., PMIDs 21949390, 28484008). Several published studies have reported analyses using 3-4 animals per group (e.g., PMIDs 28484008, 25310982, 29367382). For comparison, the study by Colombo et al. used 3-8 pups for the analysis presented in their Fig. 3. We note that the apparent variability in Fig. S2 may be accentuated by the scaling of the y-axis, which was chosen to ensure that individual data points are clearly resolved and visible.

(5) The authors stress the data of Gamage et al. (2017) on altered SC responses in DR6 mutants after injury. They employed cJun quantification to show that SC reprogramming after injury is not altered in DR6 mutants. This approach is valid and the conclusion trustworthy. Here, the addition of data showing the combined abundance of intact and degenerated myelin does not add much insight. However, Gamage et al. (2017) reported altered myelin thickness in a subset of axons at 14 days after injury, which is considerably later than the time points analyzed in the present study. While, in the Reviewer's view, the thin myelin observed by Gamage et al. in fact resembles remyelination, the authors may wish to highlight the difference in the time points analyzed.

(Response 18) We consider the additional quantification of the area occupied by intact myelin and myelin debris to provide complementary information that supports the c-Jun-based conclusion that Schwann cell injury responses are normal in DR6-deficient nerves following lesion. We agree with this reviewer that the thin myelin observed by Gamage et al. resembles remyelination, raising the possibility that axon regeneration occurred into the distal nerve stump at the studied 14d post-injury time point (see their Fig. 3). This may have been interpreted as axon protection in this study. In our study, it was impossible to examine such myelin effects since axon protection was never observed in any of the DR6 mutant models at any of the time point we investigated. We have incorporated appropriate additional text to highlight this difference. See also response #5 above.

**Reviewer #3 (Public review):**
Summary:The authors revisit the role of DR6 in axon degeneration following physical injury (Wallerian degeneration), examining both its effects on axons and its role in regulating the Schwann cell response to injury. Surprisingly, and in contrast to previous studies, they find that DR6 deletion does not delay the rate of axon degeneration after injury, suggesting that DR6 is not a mediator of this process.Overall, this is a valuable study. As the authors note, the current literature on DR6 is inconsistent, and these results provide useful new data and clarification. This work will help other researchers interpret their own data and re-evaluate studies related to DR6 and axon degeneration.Strengths:(1) The use of two independent DR6 knockout mouse models strengthens the conclusions, particularly when reporting the absence of a phenotype.(2) The focus on early time points after injury addresses a key limitation of previous studies. This approach reduces the risk of missing subtle protective phenotypes and avoids confounding results with regenerating axons at later time points after axotomy.Weaknesses:(1) The study would benefit from including an additional experimental paradigm in which DR6 deficiency is expected to have a protective effect, to increase confidence in the experimental models, and to better contextualize the findings within different pathways of axon degeneration. For example, DR6 deletion has been shown in more than one study to be partially axon protective in the NGF deprivation model in DRGs in vitro. Incorporating such an experiment could be straightforward and would strengthen the paper, especially if some of the neuroprotective effects previously reported are confirmed.

(Response 19) We thank the reviewer for these suggestions. We would like to highlight that our study addresses the role of DR6 in Wallerian degeneration, whereas *in vitro* NGF deprivation has been used to model developmental axon pruning. Previous work indicates fundamental biological differences between these regressive pathways regulating the stereotyped removal of axon segments. We feel that studying this alternative form of axon degeneration is beyond the scope of the current work and could be addressed in a separate manuscript. Although additional tests will be needed, we note that our preliminary data using samples from both DR6 knockout mouse models suggest no axon protection after NGF-deprivation in DRG neuron preparations in our hands (deprivation of the growth factor and administration of anti-NGF antibody).

(2) The quality of some figures could be improved, particularly the EM images in Figure 2. As presented, they make it difficult to discern subtle differences.

(Response 20) We have pseudocolored intact (turquoise) and degenerated (magenta) myelinated fibers on the high-resolution semithin micrographs (not electron micrographs) in the new Fig. 2 to make the distinction between the two fiber categories clearer.

**Reviewer #3 (Recommendations for the authors):**
(1) Line 121: The authors mention toluidine blue staining, but it does not appear to be shown in Figure S5.

(Response 21) This appears to be a misunderstanding. Fig. S5A shows the ultrastructure of dedifferentiated Schwann cells in transmission electron micrographs, while Figs. S5B and C show quantification of the area occupied by myelin sheaths and myelin debris profiles on osmium tetroxide and toluidine blue stained nerve sections from the two DR6 mutant models, based on semithin light microscopy. These are two different aspects of the analysis. The text has been modified in the revised manuscript to make the distinction clearer.

(2) Line 175: The authors should add NMNAT2 to the list of enzymes implicated in the regulation of Wallerian degeneration in mammals.

(Response 22) Nmnat2 and a literature reference (Milde et al., 2013) has been incorporated in the discussion of the revised manuscript to address this point.

(3) Line 201: Please correct the typo "site-by-site" to "side-by-side."

(Response 23) This typo has been corrected.